# When, Where, and What? A Benchmark for Accident Anticipation and Localization with Large Language Models

ABSTRACT

As autonomous driving systems increasingly become part of daily transportation, the ability to accurately anticipate and mitigate potential traffic accidents is paramount. Traditional accident anticipation models primarily utilizing dashcam videos are adept at predicting when an accident may occur but fall short in localizing the incident and identifying involved entities. Addressing this gap, this study introduces a novel framework that integrates Large Language Models (LLMs) to enhance predictive capabilities across multiple dimensions—what, when, and where accidents might occur. We develop an innovative chain-based attention mechanism that dynamically adjusts to prioritize high-risk elements within complex driving scenes. This mechanism is complemented by a three-stage model that processes outputs from smaller models into detailed multimodal inputs for LLMs, thus enabling a more nuanced understanding of traffic dynamics. Empirical validation on the DAD, CCD, and A3D datasets demonstrates superior performance in Average Precision (AP) and Mean Time-To-Accident (mTTA), establishing new benchmarks for accident prediction technology. Our approach not only advances the technological framework for autonomous driving safety but also enhances human-AI interaction, making the predictive insights generated by autonomous systems more intuitive and actionable.

## CCS CONCEPTS

• **Applied computing → Physical sciences and engineering**.

## KEYWORDS

Traffic Accident Anticipation; Autonomous Driving; Large Language Models; Human-AI Interaction; Dynamic Object Attention

## 1 INTRODUCTION

As autonomous driving technologies advance, the imperative to foresee and mitigate potential traffic accidents has become a cornerstone of vehicular safety strategies. Current systems primarily utilize dashcam footage to predict when and if accidents might occur. Despite substantial advancements in visual perception technologies, there remains a crucial gap in integrating these insights into autonomous systems' decision-making processes. This lack of integration restricts the systems' ability to dynamically respond to

Unpublished working draft. Not for distribution.
Permission to make digital or hard copies of all or part of this work for personal or classroom use is granted without fee provided that copies are not made or distributed for profit or commercial advantage and that copies bear this notice and the full citation on the first page. Copyrights for components of this work owned by others than the author(s) must be honored. Abstracting with credit is permitted. To copy otherwise, or republish, to post on servers or to redistribute to lists, requires prior specific permission and/or a fee. Request permissions from permissions@acm.org.
ACM MM, 2024, Melbourne, Australia
© 2024 Copyright held by the owner/author(s). Publication rights licensed to ACM.
ACM ISBN 978-x-xxxx-xxxx-x/YY/MM
https://doi.org/10.1145/nnnnnnn.nnnnnnn

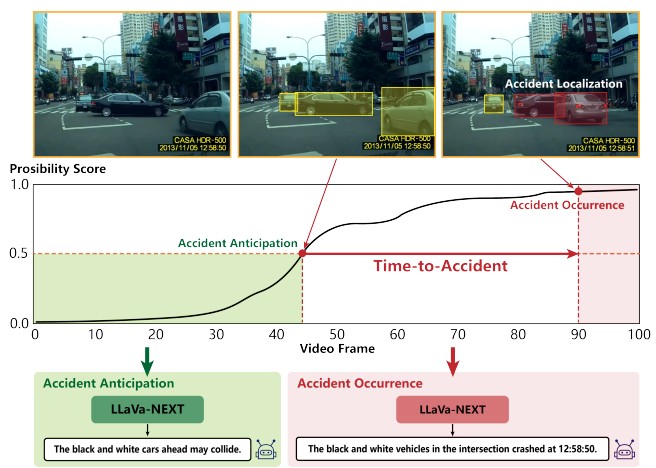

Figure 1: Illustration of accident detection, localization, and verbal warning generation performed by our model to enhance safe driving and human-AI interaction. Detected and accident-involved agents are marked as red and yellow bounding boxes, respectively.

complex driving scenarios, where not only the timing but also the location and nature of potential incidents are critical.

Traditional models often treat visual perception and decision-making as separate entities, limiting the use of rich sensory data for proactive driving adjustments. Furthermore, these models typically do not account for the dynamic nature of driving environments, failing to adapt to real-time changes and the complex interactions between various traffic participants. This static approach limits their effectiveness in the unpredictable and varied conditions typical of real-world driving. Moreover, the outputs from these models are often not translated into clear, actionable insights, reducing their practical applicability and hindering their potential to enhance safety in autonomous driving technologies.

To address these gaps, our research introduces a comprehensive framework that leverages Large Language Models (LLMs) to enhance the predictive capabilities of autonomous driving systems. By integrating cutting-edge linguistic and cognitive technologies, our approach not only predicts potential incidents more accurately but also improves the interaction between human operators and AI-driven systems, providing a richer, more intuitive user experience. Our key contributions are:

1) We have expanded the traditional scope of Accident Anticipation (What and When) to include the localization of objects involved in potential accidents (Where), a task we refer to as Accident Localization. For the first time, we utilize LLMs to analyze complex scene semantics, offering precise and timely accident alerts to passengers. Our system predicts whether an accident will occur

(What), when it might happen (When), and where it would occur (Where), thereby filling a crucial gap in accident prevention and enhancing the safety of autonomous driving.

2) We introduce a novel chain-based attention mechanism that iteratively refines feature representations through a dynamic routing mechanism enhanced by Markov chain noise models. This process allows our system to dynamically adjust attention weights across various objects within multi-agent traffic scenes, prioritizing those with higher risk levels. This attention mechanism is part of a three-stage model that preprocesses outputs from smaller models to generate multimodal inputs (image and text) for large models, guiding these large models to provide more accurate and detailed scene descriptions.

3) Our model has undergone rigorous testing on benchmark datasets such as DAD, CCD, and A3D, where it has demonstrated superior performance in key metrics like Average Precision (AP) and Mean Time-To-Accident (mTTA). The results not only surpass existing methodologies but also mark a significant advancement in accident prediction technology, setting new standards for the field.

## 2 RELATED WORK

As autonomous driving technology is gradually integrated into everyday use, ensuring its safety has become paramount. The ability of deep learning models to automatically detect or even predict accidents in advance could significantly increase confidence in autonomous driving systems. In this context, the concept of accident anticipation task was introduced in 2016 by Chan et al. [6], which builds on the foundation of accident detection to enable early prediction of accidents.

Addressing the complexities of traffic accident recognition, numerous studies [1, 15, 18, 19, 22, 29, 30, 32, 35, 39, 42, 47, 50, 52, 54, 55] have integrated Convolutional Neural Networks (CNNs) with sequence processing networks like Recurrent Neural Networks (RNNs), Long Short-Term Memory (LSTM) cells, Gated Recurrent Units (GRUs), and Graph Convolutional Networks (GCNs). This synergy enables the extraction of intricate motion patterns and temporal features from video data, facilitating the identification of potential accident precursors. Yao et al. [49] and Takimoto et al. [49] exemplify this by merging CNNs with RNNs and GRUs to analyze temporal scene dynamics and predict accidents. Basso et al. [3] introduce a CNN-based architecture for detailed vehicle behavior analysis, while Thakare et al. [41] suggest a convolutional autoencoder for feature extraction with reduced computational load, though it struggles with capturing extensive spatial patterns. Other enhancements include the adoption of attention mechanisms [2, 22–24, 43] and Transformers like UniFormerv2 [25], VideoSwin [31], and MVITv2 [11], which excel in processing visual data and understanding dynamic traffic interactions through self-attention mechanisms.

However, existing frameworks for accident anticipation and object detection often operate independently. While these models can predict accidents in advance, they fall short in identifying the participants involved in these accidents and thus lack the ability to implement appropriate actions in response to them autonomously. To address this gap, this paper extends the accident anticipation task to include the accident localization task, which predicts the occurrence of accidents in videos in advance and accurately identifies the individuals involved in the accidents.

In addition, with the rapid development of large-scale language models, more and more autonomous driving models are using multimodal large-scale models for tasks such as voice-guided driving and trajectory prediction. For example, LMDrive [38], UNIAD [17], CAVG [26], and DiLu [48], and DriveMLM [46] use multimodal sensor data, such as point clouds, combined with natural language instructions to guide vehicle navigation. GPT-Driver [33] turns trajectory planning into a language modeling task and fine-tunes GPT-3.5 accordingly. TrafficGPT [53] integrates ChatGPT with a traffic foundation model and trains on multimodal data inputs to provide comprehensive support for various traffic-related tasks. However, most existing works rely on complex multimodal inputs, which limits the range of usable datasets and complicates the creation of new datasets. In our model, we process outputs from smaller models, such as the probability of accidents occurring and information about the participants in the accidents, and use them as inputs to LLaVa-NEXT [27], thereby improving the understanding and analysis of traffic accident scenarios by the LLMs.

## 3 PROBLEM FORMULATION

This study extends the conventional scope of accident anticipation by incorporating the task of accident localization. Our objective is to devise a model that is capable of: (1) predicting the likelihood of a traffic accident occurring, (2) providing timely accident warnings if an accident is imminent, and (3) localizing the reference objects (traffic agents) involved in the accident. Given a $T$-frames dashcam video, the model is tasked with calculating a probability score $s_t$ for each frame $t \in [1, T]$, indicating the potential of an accident at that moment. An accident is predicted to occur at time step $t$ if the probability score $s_t$ first surpasses a predefined threshold $s_\theta$. We define the Time-to-Accident (TTA) as $\Delta t = \tau - t^\theta$, where $t^\theta$ is the time step when the score exceeds $s_\theta$, and $\tau$ represents the actual time step of the accident occurrence.

To localize the objects involved in accidents, we approach the task as a mapping problem: the model is required to predict the probability scores $s_t^{1:N}$ for $N$ objects in each frame $t$, aiming to pinpoint the specific objects within the video that are involved in the accident. An object, denoted as the $i$th object, is considered to be involved in an accident if $s_t^i > 0.5$; otherwise, it is not involved.

## 4 PROPOSED MODEL

Our model framework is meticulously crafted to not only anticipate accidents but also to identify objects that could precipitate such incidents, providing timely linguistic warnings for passengers. We frame the proposed model into three stages: Feature Extraction and Fusion, Accident Anticipation and Location, and Verbal Accident Alerts, as shown in Figure 2.

### 4.1 Stage-1: Feature Extraction and Fusion

In the first stage, the input dashcam video is first encoded by the MobileNetv2 [37] in the feature extractor, followed by the dual vision attention mechanism, producing a set of vision-aware features $O_V^\circ$, corresponding to each frame of the video. Concurrently, the

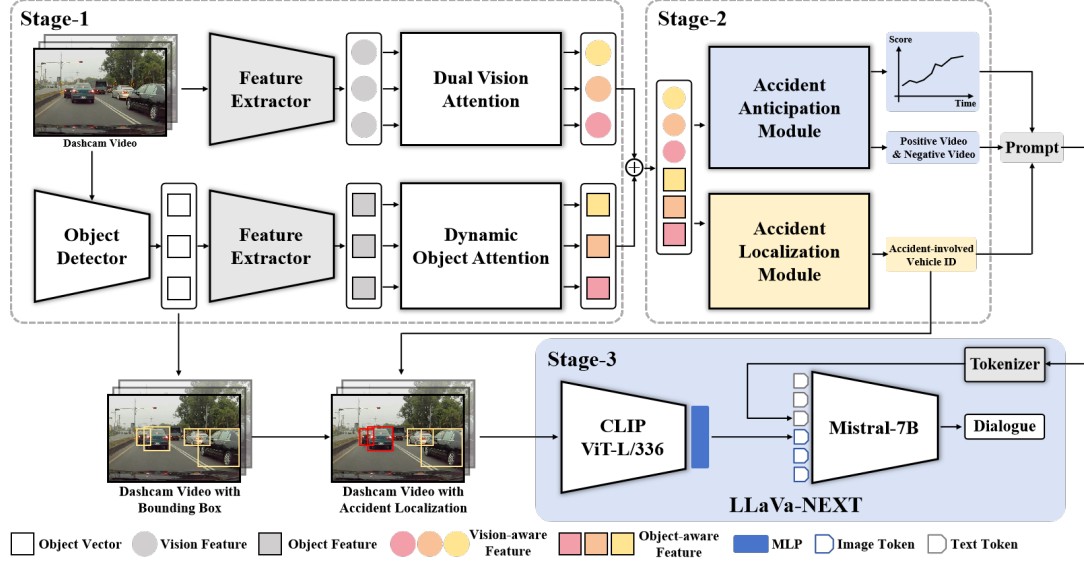

**Figure 2: The overall network architecture of our proposed model. It is a cross-modal model including three stages: Feature Extraction and Fusion, Accident Anticipation and Location, and Verbal Accident Alerts.**

raw dashcam video is also fed into the object detector to identify the object vectors $V_B = \{V_B^1, V_B^2, \ldots, V_B^T\}$ of the reference objectors via the pre-trained detector Cascade R-CNN [4]. Each vector at frame $t$, represented as $V_B^t = \{B_1^t, B_2^t, \ldots, B_N^T\}$, indicates the bounding boxes of $N$ detected objects. Next, these object vectors are refined through feature extractor and a dynamic object attention mechanism to extract precise object-aware features $O_B^\circ$.

**Dual Vision Attention.** This component is responsible for accepting the vision-aware features $O_V^\circ$. In contrast to traditional methods such as MaskFormer [8], DETR [5], and MDETR [21], which require extensive token numbers of images for self-attention and incur significant computational overhead, we introduce a dual vision attention mechanism inspired by DANet [12, 13]. As depicted in Figure. 3, it employs a "hindsight fusion" strategy. This strategy judiciously allocates attention to vision features $O_V$ extracted by the feature extractor through a two-pronged method: channel attention and position attention. Specifically, the vision features $O_V$ is converted to query $Q_P$, key $K_P$, and value $V_P$ representations via distinct convolutional layers. These representations are then utilized to generate the attention maps in the position attention, which can be represented as follows:

$$F_P = \gamma \phi_{softmax}(Q_P \times K_P^T) \times V_P + O_V \qquad (1)$$

where $\gamma$ is a trainable coefficient and $\phi_{softmax}$ denotes the softmax activation function. Furthermore, the channel attention mechanism is distinctively designed to bypass the convolutional layer embeddings typically used in position attention, favoring a direct attention approach instead. Formally,

$$F_C = \beta \phi_{softmax}(O_V \times O_V^T) \times O_V + O_V \qquad (2)$$

where $\beta$ is also a learnable coefficient. The computed channel attention maps $W_C$ along with the position attention maps $W_P$

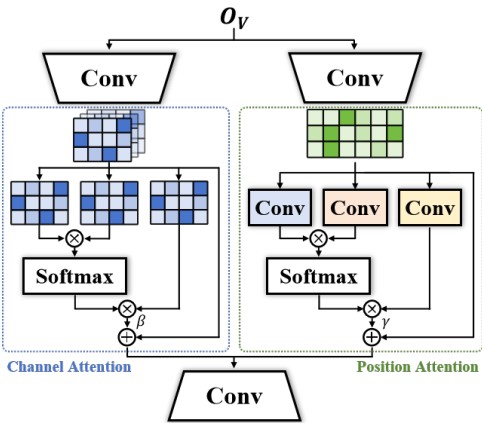

**Figure 3: Structure of the Dual Vision Attention.**

are subsequently integrated to form the refined vision-aware features $O_V^\circ = F_P \oplus F_C$. To enhance computational efficiency, we utilize down-sampling and up-sampling in conjunction with the dual vision attention mechanism. This condenses feature dimensions into a more computationally friendly latent space. This approach improves feature representation by addressing both channel and position-specific nuances while minimizing computational demands through strategic dimensional adjustments.

**Dynamic Object Attention.** The dynamic object attention mechanism is innovatively designed to dynamically adjust attention weights across various objects within multi-agent traffic scenes, effectively enabling the model to prioritize high-risk entities. Traditional attention mechanisms typically necessitate updating the attention matrix via gradient descent and backpropagation after processing a batch through the model. These approaches render the

attention matrix heavily dependent on the overall model architecture and specific hyperparameters, such as the learning rate, with feature granularity adjustments occurring across different batches.

Drawing inspiration from the capsule networks [36], we pioneer a novel chain-based attention strategy, termed dynamic diffuse attention. This mechanism fine-tunes the granularity of the feature matrix across various iterations rather than in a batch-centric manner. As illustrated in Figure. 4, the dynamic diffuse attention begins with the application of a weight matrix $W$ to effectuate a dimensional transformation on the object features $V_B$ across the $n^{th}$ iteration, $n \in [1, n]$, resulting in the enhanced object features, denoted as the $\mathbf{F}_B = W \times V_B$. Then, the embedding object features are embedded undergoes the following operation:

$$H_B^{(n)} = \phi_{softmax}(W_B^{(n)}) \cdot \phi_{dropout}(F_B) \tag{3}$$

where $W_B^{(n)}$ is a learnable weight matrix with the same shape as $\mathbf{V}_B$, and $\phi_{dropout}$ and $\cdot$ represent the application of softmax function and element-wise multiplication, respectively.

We also update the object feature representation by integrating dynamic diffuse noise. The embedded object features $H_B$ are converted via the squash activation function and then modulated by $\phi_{softmax}(W_B^{(n)})$, to which we add a level of diffuse noise $\mathcal{D}^{(n)}$ to compute the update $\Delta W_B^{(n)}$ for the weight matrix $W_B^{(n)}$:

$$\Delta W^{(n)} B = \phi_{softmax}(W_B^{(n)}) \cdot H_B^{(n)} + \mathcal{D}^{(n)} \tag{4}$$

This equation underpins the correlation between two matrices through element-wise multiplication, which improves the weighting of features with higher correlation. The noise $\mathcal{D}^{(n)}$ is intricately designed as a Markov chain $p(D^{(n)}|D^{(n-1)})$, allowing the noise from the previous iteration $\mathcal{D}^{(n-1)}$ to inform the noise in the current iteration $\mathcal{D}^{(n)}$, following the principles outlined in [16]. This stochastic approach aims to mitigate overfitting and convergence problems by progressively refining the noise through iterations:

$$\mathcal{D}^{(n)} = \sqrt{\alpha^{(n)}}\mathcal{D}^{(n-1)} + \sqrt{1 - \alpha^{(n)}}\epsilon \tag{5}$$

where $\epsilon$ denotes random Gaussian noise, introducing a measured degree of unpredictability and variance into the model. In addition, $\overline{\alpha}^{(n)} = \alpha^{(0)}\alpha^{(1)}\cdots\alpha^{(n)}$, and $\alpha^{(n)}$ is obtained as the $n$th value in a sequence generated through linear interpolation between $0.1/N$ and $20/N$ over $N$ iterations. Finally, we update the weights using $\Delta W_B : W_B^{(n+1)} = \Delta W_B^{(n)} + W^{(n)}$. The output of the $N$th iteration $W_B^N$ is the object-aware feature $O_B^{\circ}$. Notably, down-sampling and up-sampling operations are also applied in dynamic diffuse attention to reduce computational costs and further transform the feature dimensions into a latent space.

Next, we utilize a tri-layer Multilayer Perceptron (MLP) to adeptly amalgamate the vision-aware $O_V^{\circ}$ and object-aware $O_B^{\circ}$ features. This integration facilitates the generation of cross-modal features $O_C$, which serve as the input for the subsequent stage. Formally,

$$O_C^{\circ} = \phi_{MLP}(O_V^{\circ} \| O_B^{\circ}) \tag{6}$$

where the $\phi_{MLP}$ is the MLP, while $\|$ denotes matrix concatenation.

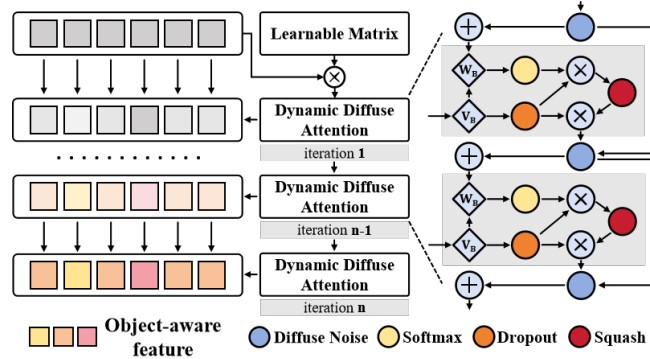

**Figure 4: Structure of the Dynamic Object Attention.**

## 4.2 Stage-2: Accident Anticipation and Location

In the second stage, we seamlessly introduce two novel modules for the task of accident anticipation and location.

**Accident Anticipation Module.** This module is architected to estimate in real-time the probability scores $S = \{S^1, S^2, \dots, S^T\}$ for each frame of the input video. This estimation serves to identify the likelihood of an accident occurring, thereby facilitating the earliest possible detection and providing a critical lead time for preventive action. To achieve this, we employ GRUs and MLPs to refine the cross-modal features $O_C^{\circ}$ synthesized during the first stage of our framework. Subsequently, we implement a series of three convolution-deconvolution operations across varying receptive fields. This approach ensures the assimilation of temporal dependency over diverse scales, culminating in a nuanced and precise prediction of accident probability for any given frame of the video.

**Accident Localization Module.** In this module, we utilize a sophisticated attention mechanism in conjunction with a GRU to compute the probability values of accident occurrence for each detected object. To ensure coherent reasoning, we harmonise the vision-aware $O_V^{\circ}$, object-aware $O_B^{\circ}$, and cross-modal $O_C^{\circ}$ features by projecting them onto the same semantic space through linear projection and L2 normalisation. This projection yields the query $Q^{\circ}$, key $K^{\circ}$ and value $V^{\circ}$ representations, formalised as follows:

$$Q^{\circ} = W_Q^{\circ}\phi_{MLP}(O_V^{\circ}), K^{\circ} = W_K^{\circ}\phi_{MLP}(O_B^{\circ}), V^{\circ} = W_V^{\circ}\phi_{MLP}(O_C^{\circ}) \tag{7}$$

where $W_Q^{\circ}, W_K^{\circ}, W_V^{\circ}$ represent learnable matrices tailored for linear projection. The transformed query $Q^{\circ}$, key $K^{\circ}$, and value $V^{\circ}$ are then fed into the attention block, articulated as follows:

$$F_c = \phi_{softmax}(\frac{Q^{\circ} \cdot K^{\circ}}{\sqrt{d_k}}) \cdot V^{\circ} \tag{8}$$

where $d_k$ denotes the dimension of the transformed vectors. The attention-derived matrix $F_c$ is further refined by a GRU. This GRU uses scatter and gather operations to efficiently parallelize the acquisition of contextual information alongside the learning of spatio-temporal interdependencies between agents. This innovative approach enhances the model's ability to capture and analyse the complex dynamics present in multi-agent traffic scenes.

Subsequently, a softmax function calculates likelihood scores for each detected object. This crucial step allows the identification of

the top-$k$ objects that have the highest association with potential accidents. By prioritizing these objects, our model focuses on the most critical elements within the traffic scenes and significantly enhances its practical utility by providing actionable insights for accident prevention.

### 4.3 Stage-3: Verbal Accident Alerts

Recent studies [14, 26] have highlighted the importance of natural language commands in improving passenger experience and acceptance of autonomous vehicles (AVs). Therefore, beyond the critical functions of accident anticipation and localization, our model framework endeavors to enhance human-AI interaction by providing verbal accident alerts to passengers.

This stage is intricately designed to deliver precise and timely traffic accident warnings, utilizing the latest Large Language Model (LLM), LLaVa-NEXT [27, 28]. It processes dashcam video footage, coupled with accident localization data and structured prompts [44] as inputs. These prompts encompass exhaustive scene semantic annotations derived from the second stage—such as probability scores and Time-to-Accident (TTA) derived from stage two outputs—to guide the model to fully understand complex semantic scenes.

To prepare the input for the Mistral-7B model [20], we use CLIP [34] and Vision Transformer (ViT) [10] for initial object recognition and image tokenization within the video. This process identifies key entities such as traffic signs, vehicles, and pedestrians, thereby enriching the visual cues available to the model. At the same time, the input prompts are tokenized into sequences using the Bidirectional Transformer (BERT) model's WordPieces tokenizer [9], and then also integrated into the Mistral-7B model. Finally, the Mistral model synthesizes this multimodal information and generates dialogues that articulate the expected timing of the accident and the specific accident-involved traffic agents. To the best of our knowledge, we are the first to leverage the linguistic prowess of the LLMs to produce accident alert dialogues. This innovation fills a crucial gap in the realm of safe driving and human-machine interaction, marking a significant step forward in the integration of linguistic capabilities into autonomous driving technologies.

## 5 TRAINING

Our training loss function consists of three main components: the score loss $L_S$ for predicting the probability scores of all frames in dashcam videos, the anticipation loss $L_A$ for predicting whether accidents occur in dashcam videos, and the localization loss $L_M$ for locating vehicles involved in accidents.

The score loss $L_S$ is calculated using the ground-truth accident time $\tau$ and the probability scores $s^{n,t}$ at time step $t$. Specifically, given the positive videos (i.e., videos with accidents), we set the probability scores $s^{p,t}$ of each frame to be close to 1, while for negative videos (i.e., videos without accidents), we set these scores $s^{n,t}$ to approach 0. To account for the increasing relevance of frames closer to the accident time, we introduce a weighting coefficient $e^{-\max\left(\frac{\tau-t}{\lambda}, 0\right)}$ that penalizes probability scores closer to the accident time $\tau$, where $\lambda$ is a decay factor set to 20. For positive and negative videos, the labels $\mathcal{L}_S^p$ and $\mathcal{L}_S^n$ are set to 1 and 0, respectively,

resulting in the following formulation for $L_S$:

$$L_S = \frac{1}{V} \frac{1}{T} \sum_{v=1}^{V} \sum_{t=1}^{T} e^{-\max\left(\frac{\tau-t}{\lambda}, 0\right)} \left[ -\mathcal{L}_S^p \log(s^{p,t}) - (1 - \mathcal{L}_S^n) \log(1 - s^{n,t}) \right] \tag{9}$$

Furthermore, the anticipation loss $L_A$ can be defined as follows:

$$L_A = \frac{1}{V} \sum_{v=1}^{V} \left[ -\mathcal{L}_A^p \log(l_a) - (1 - \mathcal{L}_A^p) \log(1 - l_a) \right] \tag{10}$$

where $l_a$ is the output of the accident anticipation module, and $V$ is the number of dashcam videos. We assign a label $\mathcal{L}_A^p = 1$, indicating a positive instance. Conversely, for videos devoid of accidents, we denote these as negative instances with a label $\mathcal{L}_A^n = 0$.

In addition, the localization loss $L_M$ is specifically designed to instruct the model in discerning whether each detected object within the video plays a role in an accident. For every object $n \in [1, N]$ that appears in the video, we define labels for objects positively associated with an accident ($L_M^p = 1$) and those negatively associated ($L_M^n = 0$). Consequently, the localization loss, $L_M$, is formulated as follows:

$$L_M = \frac{1}{V} \frac{1}{T} \frac{1}{N} \sum_{v=1}^{V} \sum_{t=1}^{T} \sum_{n=1}^{N} \left[ -L_M^{p,t} \log(l_m^{t,n}) - (1 - L_M^{p,t}) \log(1 - l_m^{t,n}) \right] \tag{11}$$

Here, $l_m^{t,n}$ represents the predictive output for the $n^{th}$ object at frame $t$, with $T$ signifying the total frame count. This loss function enhances the model's capability in accurately determining the involvement of each detected object in potential accident scenarios across all frames, thereby optimizing the accuracy of accident localization.

During the first training phase, the final loss function $L$ is the sum of score loss $L_S$ and anticipation loss $L_A$, i.e., $L = L_S + \eta L_A$, where $\eta$ is a constant coefficient. In the second training phase, the loss function $L$ consists only of $L_M$. This structured approach allows for a nuanced and effective model training strategy that addresses the complexities of traffic accident detection and localization in dashcam video.

## 6 EXPERIMENT

### 6.1 Datasets

**DAD.** The Dashcam Accident Dataset (DAD) [6] compiles a collection of 620 dashcam recordings from six prominent cities in Taiwan, each lasting 5 seconds and captured at a rate of 20 frames per second. From these recordings, 1750 video segments were extracted, including 620 accident segments and 1130 non-crash segments. For the segments with accidents, the collision time was set to the 90th frame. Among the three datasets discussed, the DAD dataset is the only one that includes annotations for object detection bounding boxes, object IDs, object categories, and labels indicating the occurrence of accidents. This unique composition makes the DAD dataset particularly suitable for tasks related to the localization of objects involved in accidents. The segmentation of the dataset for model training and evaluation purposes allocates 70% of the data to the training set, which is further divided into 455 accident and 829 non-accident segments, while the test set contains 165 accident and 301 non-accident segments.

**CCD.** The Car Crash Dataset (CCD) [1], is an extensive collection of 4500 video recordings annotated with different environmental conditions (day/night, different weather conditions such as snow, rain, or clear sky), the involvement of bicycles and pedestrians, and detailed explanations of the causes of the accidents. Each video, which captures 5 seconds of footage at a playback rate of 10 frames per second, marks accidents in positive cases at the 40th frame. This dataset is strategically divided into training (80%) and test (20%) sets, maintaining a balance of one positive to two negative videos in both segments.

**A3D.** The AnAn Accident Detection (A3D) dataset [49], contains 1500 dashcam video clips from different East Asian urban environments, representing a range of weather conditions and times of day. These clips are each 5 seconds long, with a frame rate of 20 frames per second achieved through down-sampling. In the dataset, accidents within positive video segments are identified at the 80th frame. The split of the data for development purposes is set at 80% training and 20% testing.

## 6.2 Metrics

In the area of traffic anticipation and localization tasks, three primary evaluation metrics are used: Average Precision (AP), Mean Time-To-Accident (mTTA), and Accident Object Localization Accuracy (AOLA).

**Average Precision (AP).** AP serves as a measure to evaluate the model's ability to accurately detect the occurrence of traffic accidents within videos, especially in scenarios where there is an imbalance between positive and negative samples. In binary classification tasks, assuming that $TP$, $FP$, and $FN$ represent the number of true positives, false positives, and false negatives, respectively, we can calculate the model's recall $R = \frac{TP}{TP+FN}$ and precision $P = \frac{TP}{TP+FP}$. Recall indicates the proportion of positive instances that are correctly predicted, while precision reflects the proportion of positive predictions that are actually positive. A precision-recall curve is plotted from these values, and AP is defined as the area enclosed by this curve and the coordinate axes. In practice, the area under the curve is approximated by discrete summation:

$$AP = \int P(R)dR = \sum_{k=0}^{m} P(k)\Delta R(k) \tag{12}$$

**Mean Time-To-Accident (mTTA).** mTTA quantifies the ability of the model to predict in advance the occurrence of an accident among the positive samples. If an accident occurs at frame $\tau$, TTA is defined as $\Delta t = \tau - t_\theta$, where $t_\theta$ satisfies $s_t \geq s_\theta$ for $t \geq t_\theta$ and $s_t < s_\theta$ for $t < t_\theta$, where $s_\theta$ represents the threshold for the accident probability score. Across all possible thresholds $s_\theta \in [0, 1]$, mTTA is the average of all TTAs, i.e., $mTTA = \frac{1}{n}\sum_{s_\theta} TTA$.

**Accident Object Localization Accuracy (AOLA).** AOLA assesses the accuracy of the model in predicting the occurrence of accidents for all detected objects. For a total of $N_V$ videos, each containing $f$ frames and $N_O$ objects per frame, AOLA is defined as follows:

$$AOLA = \frac{\sum_{i=1}^{N_V} \sum_{j=1}^{f} n_o}{\sum_{i=1}^{N_V} \sum_{j=1}^{f} N_O} \tag{13}$$

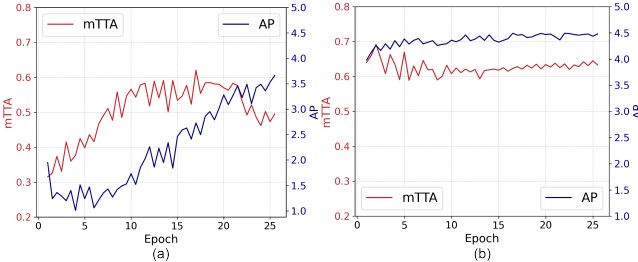

**Figure 5: Comparative analysis of trends in AP and mTTA metrics during training between DSTA (a) and our proposed model (b). The performance metrics are recorded at every half-epoch interval.**

**Table 1: Comparison of models seeking balance between mTTA and AP on three datasets. Bold and underlined values represent the best and second-best performance in each category. Instances where values are not available are marked with a dash ("-").**

| Model | DAD | | | CCD | | A3D | |
|---|---|---|---|---|---|---|---|
| | AP(%)↑ | mTTA(s)↑ | AOLA↑ | AP(%)↑ | mTTA(s)↑ | AP(%)↑ | mTTA(s)↑ |
| DSA [7] | 48.1 | 1.34 | - | 98.7 | 3.08 | 92.3 | 2.95 |
| ACRA [51] | 51.4 | 3.01 | - | 98.9 | 3.32 | - | - |
| AdaLEA [40] | 52.3 | 3.43 | - | 99.2 | 3.45 | 92.9 | 3.16 |
| UString [1] | 53.7 | 3.53 | - | 99.5 | 3.74 | 93.2 | 3.24 |
| DSTA [23] | 56.1 | 3.66 | - | 99.6 | 3.87 | 93.5 | 2.87 |
| GSC [45] | 60.4 | 2.55 | - | 99.4 | 3.68 | 94.9 | 2.62 |
| **Ours** | **69.2** | **4.26** | 0.89 | **99.7** | **3.93** | **96.4** | **3.48** |

where $n_o$ denotes the number of correctly predicted objects per frame.

## 6.3 Implementation Details

In this study, Pytorch is used for the implementation, and training and testing are performed on an A40 48G GPU. For the pre-trained model, we use MobileNetv2, from which 1280 feature dimensions are extracted. For the model hyperparameters, we set the number of dynamic routing iterations within the Dynamic Object Attention mechanism to 8, with a maximum of 19 objects detected per frame. For the loss function parameters, we set a decay coefficient $\lambda = 20$ and a loss function ratio coefficient $\eta = 10$. For the training parameters, we set the model learning rate to $1 \times 10^{-4}$, with a batch size of 16. We use ReduceLROnPlateau as the learning rate scheduler to ensure that each model is trained for at least 10 epochs. See **Appendix** for more implementation details.

## 6.4 Comparison to State-of-the-art (SOTA)

We conduct extensive experiments on the DAD, CCD, and A3D datasets. Our model demonstrates superior performance in both AP and mTTA metrics, as detailed in Table 1. Notably, on the DAD dataset, our model achieved a remarkable 14.6% improvement in AP and a 16.4% increase in mTTA compared to the second-performing model. While enhancements on the CCD and A3D datasets were

**Table 2: Comparison of models for the best AP on DAD datasets. TTA@80 means the value of mTTA at recall equals to 80%. Bold and underlined values represent the best and second-best performance of each category. Instances where values are not available are marked with a dash ("-").**

| Model | Backbone | Publication | AP(%)↑ | mTTA(s)↑ | TTA@R80(s)↑ |
|---|---|---|---|---|---|
| ACRA[51] | VGG-16 | ACCV'16 | 51.40 | - | - |
| DSA [7] | VGG-16 | ACCV'16 | 63.50 | 1.67 | 1.85 |
| UniFormerv2 [25] | Transformer | ICCV'23 | 65.24 | - | - |
| VideoSwin [31] | Transformer | CVPR'22 | 65.45 | - | - |
| MVITv2 [11] | Transformer | CVPR'21 | 65.45 | - | - |
| DSTA [23] | VGG-16 | TITS'22 | 66.70 | 1.52 | 2.39 |
| UString [1] | VGG-16 | ACMMM'20 | 68.40 | 1.63 | 2.18 |
| GSC [45] | VGG-16 | TIV'23 | 68.90 | 1.33 | 2.14 |
| **Ours** | MobileNetv2 | - | **69.20** | **4.26** | **4.33** |

**Table 3: Ablation studies of different modules on DAD dataset. DIA, DOA, AAM, and ALM represent Dual Vision Attention, Dynamic Object Attention, Accident Anticipation Module, and Accident Localization Module, respectively.**

| Model | Component | | | | Evaluation Metric | | |
|---|---|---|---|---|---|---|---|
| | DIA | DOA | AAM | ALM | AP(%)↑ | mTTA(s)↑ | AOLA↑ |
| A | ✘ | ✔ | ✔ | ✔ | 61.4 | 4.17 | 0.81 |
| B | ✔ | ✘ | ✔ | ✔ | 56.8 | 3.69 | 0.72 |
| C | ✔ | ✔ | ✘ | ✔ | 65.3 | 2.46 | 0.86 |
| D | ✔ | ✔ | ✔ | ✘ | 59.5 | 4.01 | 0.65 |
| original | ✔ | ✔ | ✔ | ✔ | 69.2 | 4.26 | 0.89 |

more modest, this can be attributed to the already near-optimal performance of competing models on these datasets. Additionally, as indicated in Table 2, our model secured the top scores across both AP and mTTA metrics. Our analysis revealed that, unlike competing models which faced challenges in optimizing the trade-off between AP and mTTA, our model adeptly maintains this balance throughout the training process. Table 5 illustrates that while other models, such as DSTA, peaked in AP at the 20th epoch before experiencing a rapid decline, our model reaches peak performance by the 2nd epoch and maintains a minimal decline in performance thereafter, highlighting its rapid convergence and resilience to overfitting.

Furthermore, our model undergoes rigorous multi-class accuracy (AOLA) testing on the DAD dataset, achieving an accuracy rate of nearly 90%. This test involves classifying each video frame into one of 19 possible object categories, demonstrating the model's accuracy in recognizing and classifying a wide range of objects in complex traffic scenes. Achieving such a high accuracy rate, especially in a multi-class setting, underscores the effectiveness and adaptability of our model and sets a new benchmark in accident anticipation and localization for autonomous driving systems.

## 6.5    Ablation Studies

**Ablation Studies of Different Components.** Table 3 presents our ablation study for four key components: dual vision attention, dynamic object attention, accident anticipation module, and accident localisation module, highlighting their indispensability within our model framework. Model A, lacking dual vision attention, shows

**Table 4: Ablation studies of the Dynamic Object Attention on iterations. Num-iteration means the number of iterations that Dynamic Route used during the training and testing process. TC means the time consumption during training. During the training process, the time consumption by the model with Num-iteration=1 is set as a baseline of 1.**

| Index | Num-iteration | | Evaluation Metrics | | | |
|---|---|---|---|---|---|---|
| | Train | Test | AP(%)↑ | mTTA(s)↑ | AOLA↑ | TC(%)↓ |
| 1 | 2 | 2 | 63.1 | 3.95 | 0.82 | 1.02 |
| 2 | 4 | 4 | 66.8 | 4.10 | 0.85 | 1.04 |
| 3 | 6 | 6 | 69.2 | 4.26 | 0.89 | 1.07 |
| 4 | 8 | 8 | 68.7 | 4.28 | 0.88 | 1.12 |
| 5 | 10 | 10 | 67.4 | 4.23 | 0.86 | 1.15 |
| 6 | 6 | 1 | 66.4 | 4.16 | 0.82 | - |
| 7 | 6 | 2 | 67.1 | 4.20 | 0.85 | - |
| 8 | 6 | 3 | 68.3 | 4.22 | 0.87 | - |
| 9 | 6 | 4 | 68.9 | 4.23 | 0.88 | - |
| 10 | 6 | 5 | 69.0 | 4.25 | 0.88 | - |
| 11 | 6 | 6 | 69.2 | 4.26 | 0.89 | - |

decreases in AP, mTTA and AOLA metrics, highlighting the importance of incorporating learnable attention weights in global image processing. Model B, devoid of dynamic object attention, experiences a significant decrease in all three metrics due to the absence of key object features, further highlighting the importance of computing fine-grained correlations between detected objects to focus the model on accident-relevant traffic agents for more accurate anticipation. Furthermore, Model C, which omits the output of probability scores and focuses solely on binary accident prediction, maintains its AP score but experiences reduced performance in mTTA. Finally, Model D, which excludes the accident localization module, results in a significant decrease in the AOLA metric and a decrease in both AP and mTTA scores. This indicates that the prediction of accident-involved traffic agents not only improves the model's accuracy (AP), but also its timeliness (mTTA). In summary, the results of these ablation studies confirm the effectiveness of each model component. Together, these components synergistically perform the tasks of accident anticipation and localisation with improved accuracy and timeliness. See **Appendix** for more details on ablation studies.

**Ablation Studies of Dynamic Object Attention.** This study introduces the dynamic object attention mechanism that leverages noise generated by a Markov chain of diffusion model. Through multiple iterations, this mechanism progressively learns the correlations between different detected entities and iteratively updates their feature representations. To validate the importance of multi-layer iterations and the efficacy of incorporating diffusion noise, we conduct a series of ablation experiments. As illustrated in Table 4, Experiments 1-5 demonstrate that the model achieves optimal Average Precision (AP) and ALOA metrics when the number of iterations, Num-iteration, is set to 6. An increase or decrease in the number of iterations respectively leads to overfitting or underfitting. Furthermore, the duration of the process does not significantly increase with additional iterations, making Num-iteration=6 the optimal choice. Experiments 6-11 investigate the impact of varying the number of test iterations while maintaining the same number

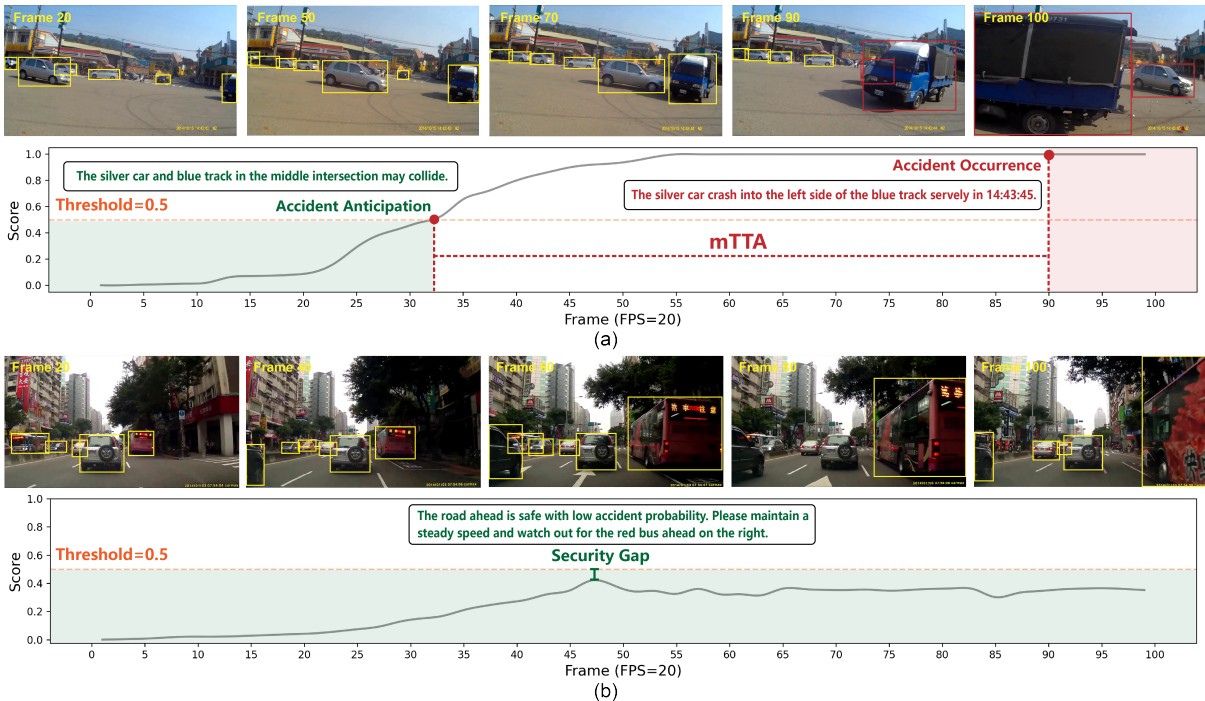

Figure 6: Visualization of Model Performance on the DAD dataset.

of training iterations. The results indicate that model performance is not significantly affected by reducing the number of test iterations. This is due to the shared weight parameters across iterations, which maintain effectiveness even with significantly fewer test iterations than in training. In addition, Table 5 further compares the performance with and without the use of different types of noise. Experiments 1-3 indicate that noise introduction enhances model generalization; however, excessive noise (Experiment 3) degrades performance. Experiments 4-5 show that linking noise across iterations significantly improves outcomes, with Markov chain-based connections proving most effective. In summary, these ablation study results highlight the importance of multilayer iterations and the strategic inclusion of diffusion noise in improving model accuracy and generalization.

## 6.6 Visualization

Figure 6 shows the temporal variation of the output probability scores indicating the likelihood of an accident. As shown in Figure 6 (a), our model successfully identifies vehicles involved in accidents (indicated by red bounding boxes) and outputs probability scores close to 1 after the accident. Prior to the accident, the model's predicted probability scores exceed the threshold early, suggesting that the model can detect changes in the target agents' behavior in the image and infer the increasing likelihood of an accident under continuing conditions, thus assigning higher probability scores in anticipation. Conversely, as shown in Figure 6 (b), the model's predictions do not exceed the threshold, indicating that no accident has occurred in the video.

Table 5: Ablation studies of the dynamic object attention on noise. "None" indicates no noise is applied, while "Same" indicates using identical Gaussian noise for each iteration loop. "Different" indicates using different Gaussian noise for different iteration loops. "Linear" denotes using a simple linear relationship between the Gaussian noise across loops. "Markov chain" describes the method used in this study.

| Index | Noise | Evaluation Metrics | | |
| --- | --- | --- | --- | --- |
| | | AP(%)↑ | mTTA(s)↑ | AOLA↑ |
| 1 | None | 63.6 | 4.01 | 0.82 |
| 2 | Same | 64.3 | 4.09 | 0.84 |
| 3 | Different | 63.8 | 3.86 | 0.81 |
| 4 | Linear | 67.7 | 4.33 | 0.87 |
| 5 | Markov chain | 69.2 | 4.26 | 0.89 |

## 7 CONCLUSION

In this study, we extend accident anticipation to accident localization by using LLMs for detailed scene analysis, enabling precise accident warnings about what, when, and where of potential incidents, thus significantly improving driving safety. We also present a novel three-stage model tailored to the task of traffic anticipation and localization. It introduces a novel attention mechanism that dynamically refines feature representations, prioritizing high-risk objects in traffic scenes. Moreover, we are the first to apply the LLMs to generate verbal accident alerts in accident anticipation, significantly enhancing human-AI interaction. Our proposed model showcases superior performance on key metrics in real-world datasets such as DAD, CCD, and A3D, setting a new benchmark in this field.

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
