# OpenReview forum: "When, Where, and What? A Benchmark for Accident Anticipation and Localization with Large Language Models"
_acmmm.org/ACMMM/2024/Conference — MM2024 Oral_

### Official Review · Reviewer_sPkx · 2024-05-24

**Rating:** 4
**Confidence:** 3

**Summary:**

This work proposes an approach for accident anticipation in autonomous driving systems. It introduces using LLMs to enhance the model's capabilities in predicting when, where, and when the accident would occur. The proposed approach anticipates accidents and also improves the interaction between the driver and the driving system by providing a better user experience. The proposed approach consists of a novel chain-based attention mechanism to refine feature representations, which enables the model to focus on different objects that could be involved in the accident. The model consists of three stages: Feature extraction and Fusion, Accident Anticipation and Localization, and Verbal alerts to the driver.  For evaluation of the model, datasets DAD, CCD, and A3D are used and the results show that the proposed model has improved over the state-of-the-art on the metrics Average Precision (AP) and Mean Time-To-Accident (mTTA).

**Strengths:**

--> The paper is well-written and well-organized which makes it easy to read.
--> Experimental evaluation on multiple datasets with improvements over the current SOTA is a definite positive of the paper.
--> Also ablation experiments show the contribution of the individual components and help understand the design decisions.
--> Figures showing the architecture of the different modules help in an easy understanding of the proposed approach.

**Limitations:**

--> Fig. 1 caption, should it not be "yellow and red.." instead of "red and yellow.."?
--> It is not clear how the outputs from the Accident Anticipation Module and Accident Localization Module are converted to text prompts that could be passed as input to the LLava-Next text model (Mistral-7B)
--> My understanding is that here the use of LLava-Next is only to generate a textual response (dialogue) so that it can be relayed to the driver via some text-to-speech model. It seems to be an overkill to use a heavy LLava-Next model for this task. Also, there is no evaluation of the dialogues generated by the model.
--> The results shown in the paper only evaluate the accident anticipation but not the localization output. How is localization contributing to improving the anticipation results?
--> Results in Table 1 on CCD for the model DSTA are wrongly reported. Results from the DSTA paper [1], show a score of 4.87 and not 3.87.

**Suitability:**

2

---

### Official Review · Reviewer_NStK · 2024-05-24

**Rating:** 5
**Confidence:** 2

**Summary:**

This work tackles the research problem of Accident Anticipation and Localization in driving scenes. Unlike previous works focusing on accident anticipation (When and What), this work expands the task with the localization of potential accidents (Where). Precisely, the proposed method performs respectively image feature extraction, object detection, accident anticipation, and accident localization. The outputs of them are used to form the prompt that is used to query a finetuned Large Language Model. The LLM summarizes and decides if there's a potential accident and where it could happen.
The proposed method achieves state-of-the-art performance in several benchmarks such as DAD, CCD and A3D, more importantly it extends the accident anticipation task to accident localization.

**Strengths:**

- The extension of accident anticipation to more fine-grained accident anticipation and localization
- Leveraging state-of-the-art LLM (Mistral-7B) and VLM (CLIP) to provide more precise and interactive accident anticipation for driving.
- The careful design and detailed ablation validate the effectiveness of the proposed method
- The state-of-the-art performance in several benchmarks such as DAD, CCD and A3D

**Limitations:**

- The use of language model is not the first time in driving scenarios, works such as [1, 2]. A discussion with similar applications of LLMs could be provided.
- I have concerns about the inference speed of the proposed complex pipeline since it is critical for anticipating accidents on time.
-  The proposed pipeline uses an off-the-shelf object detector to locate the objects and the rest of the modules seem dependent on the object detection quality. A discussion on the possible miss detection of objects and its impact on the LLM decision is interesting to have.
[1] Driving with LLMs: Fusing Object-Level Vector Modality for Explainable Autonomous Driving
[2] DriveGPT4: Interpretable End-to-end Autonomous Driving via Large Language Model

**Suitability:**

2

---

### Official Review · Reviewer_tqTc · 2024-05-27

**Rating:** 4
**Confidence:** 2

**Summary:**

This paper focuses on predicting accidents in vehicle driving. It proposes novel attention designs, an accident localization module and uses LLM  for human text generation. First, videos are sent to the encoder for feature extraction. At the same time, another parallel branch is used for object detection. Then, video features and object instance features are concatenated and sent to the accident anticipation/localization modules. The output prompts are sent to the LLM. Based on the accident analysis prompts and object detection results, the MLLM can output texts for accident analysis results.

**Strengths:**

This paper is clear and easy to follow. The motivation is clear and convincing, the methodology is novel with sufficient details, and the experiments are solid. The application of LLM here is reasonable and can be seen as a good application of MLLM. The reviewer is satisfied with this paper.

**Limitations:**

I am not an expert in this specific topic, so I do not have too many concerns about this paper. But there are still some minor issues:

1. The authors can provide the efficiency comparison of methods. Considering this is for vehicle driving purpose, an online system is preferred.

2. The authors can cite more papers on MLLM for autonomous driving in the related work section.
[1] Xu, Zhenhua, et al. "Drivegpt4: Interpretable end-to-end autonomous driving via large language model." arXiv preprint arXiv:2310.01412 (2023).
[2] Ma, Yingzi, et al. "Dolphins: Multimodal language model for driving." arXiv preprint arXiv:2312.00438 (2023).

**Suitability:**

3

---

### Official Review · Reviewer_WdSo · 2024-05-28

**Rating:** 4
**Confidence:** 2

**Summary:**

This paper addresses the critical need for enhanced accident anticipation in autonomous driving systems. Traditional models predict the timing of accidents but often fail to localize incidents and identify involved entities. To overcome these limitations, the authors propose a novel framework that integrates Large Language Models (LLMs) to improve the predictive capabilities regarding what, when, and where accidents might occur. The framework demonstrates superior performance on benchmark datasets (DAD, CCD, and A3D) in terms of Average Precision (AP) and Mean Time-To-Accident (mTTA), setting new standards in accident prediction technology.

**Strengths:**

1. The integration of LLMs for accident prediction and localization represents a holistic approach to enhancing autonomous driving safety, addressing both temporal and spatial aspects of accident anticipation.
2. The chain-based attention mechanism and the dynamic object attention mechanism show significant advancements in prioritizing high-risk elements and refining feature representations.
3.  Rigorous testing on multiple benchmark datasets demonstrates the effectiveness and robustness of the proposed framework, establishing new performance benchmarks.

**Limitations:**

1. The figures in your paper are a bit blurry, such as Figure 3 and Figure 4. Please consider replacing them with clearer ones.
2. The methods section does not provide enough detail on key aspects. For instance, more detailed information about the prompt generation after stage 2  would be beneficial.
3. Certain critical terms such as "channel attention" and "position attention" are not adequately defined, which can hinder the reader's understanding of the paper's contributions and methodologies. And the motivation of this **Dual Vision Attention** module is not clear. The authors need to highlight its necessity. Besides, the visualization of the proposed attention mechanism would be beneficial.
4. The definition of the mathematical symbols that appear in the article is not clear, such as the shape of matrix $O_{V}^o$, $O_{B}^o$ , which makes it difficult for the reader to understand the details of each module.

**Suitability:**

3

---

### Meta-Review · Area_Chair_qVXU · 2024-06-23

**Recommendation:** Accept (Oral)
**Confidence:** 5

**Metareview:**

The paper received four positive reviews out of four. I am therefore recommending to accept the paper.